# Patterns and Mitigation Strategies for Rejected Claims Among Health Facilities Providing Services for the National Health Insurance Fund in Mwanza, Tanzania

**DOI:** 10.3390/healthcare13030320

**Published:** 2025-02-04

**Authors:** Ritha Fulla, Namanya Basinda, Theckla Tupa, Peter Chilipweli, Anthony Kapesa, Eveline T. Konje, Domenica Morona, Stephen E. Mshana

**Affiliations:** 1Bugando Medical Centre, Catholic University of Health and Allied Sciences, Mwanza P.O. Box 1464, Tanzania; ritharenerfulla@gmail.com (R.F.); thecklatupa1@gmail.com (T.T.); 2Department of Community Medicine, School of Public Health, Catholic University of Health and Allied Sciences, Mwanza P.O. Box 1464, Tanzania; chilipwelipeter64@gmail.com (P.C.); anthony1kapesa@gmail.com (A.K.); 3Department of Epidemiology and Biostatistics, Catholic University of Health and Allied Sciences, Mwanza P.O. Box 1464, Tanzania; ekonje28@bugando.ac.tz; 4 Department of Parasitology and Entomology, Weill-Bugando School of Medicine, Catholic University of Health and Allied Sciences, Mwanza P.O. Box 1464, Tanzania; dmorona@bugando.ac.tz; 5Department of Microbiology and Immunology, Weill-Bugando School of Medicine, Catholic University of Health and Allied Sciences, Mwanza P.O. Box 1464, Tanzania; mshana72@bugando.ac.tz

**Keywords:** rejected claims, health insurance, health insurance rejected claims, pattern of rejected claims, mitigation strategy for rejected claims

## Abstract

**Background**: Rejected medical claims pose a significant challenge for healthcare facilities accredited by Tanzania’s National Health Insurance Fund (NHIF). Despite the NHIF’s role in reducing out-of-pocket costs, claim rejections have been a persistent issue, largely due to documentation errors, coding mistakes, and non-compliance with NHIF regulations. This study determined the patterns of rejected claims and the strategies employed by NHIF-accredited hospitals to mitigate these challenges. **Methodology**: This cross-sectional study was conducted between July and August 2024 and used quantitative and qualitative approaches. The study utilized secondary data (August 2023 to January 2024) on the rejected claims from 46 healthcare facilities (HFs) and key informant interviews from the respective selected facilities. Descriptive data analysis was carried out using STATA version 15 and qualitative data analysis was conducted using NViVo2 version 12 software. **Results**: A total of 46 public (27) and private (19) HFs were included in this study. The data revealed significant variation in the average number of items rejected per claim across HFs, ranging from 0.21 in a regional referral hospital to 1.21 in a zonal hospital. Non-adherence to standard treatment guidelines (STGs) was significantly more common (*p* < 0.001) in polyclinics, accounting for 17.2% of the items rejected, and with the lowest number (0.8%) seen in zonal hospitals. Overutilization (drugs and investigations) was commonly reported in all HFs, ranging from 12.5% in polyclinics to 31.8% in district hospitals (*p* < 0.001). Non-applicable consultation charges were only reported in one zonal hospital. To mitigate these rejections, HFs implemented strategies such as immediate error verification, regular communication with NHIF, staff training, technology use, and regular supervision by the internal audit units. Despite these efforts, challenges persisted, particularly those stemming from complex NHIF policies, which account for most rejections in zonal health facilities. **Conclusions**: There are significant variations in rejection patterns among HFs, with attendance date anomalies, non-adherence to STGs, NHIF pricing, and overutilization being the most common reasons across all HFs. Strategies to address rejections should be tailored to specific health facilities, coupled with electronic systems that will detect errors during patient management.

## 1. Introduction

Health insurance is one of the main financing mechanisms currently being used in low- and middle-income countries to improve access to quality health services [1,2,3]. With the goals of improving access to high-quality healthcare services and offering financial security, Tanzania’s National Health Insurance Fund (NHIF) is an essential part of the country’s healthcare system [2,4,5,6]. The NHIF has helped Tanzania reduce out-of-pocket costs, which are a roadblock to adopting universal health coverage. Furthermore, the NHIF has been key in improving the healthcare system in Tanzania [2,3]. However, reports indicate that claim rejections have been a major problem for NHIF-accredited healthcare facilities (HFs) [2,4,5,6].

Rejected medical claims are defined as submitted claims that are rejected for payment by insurance companies [1]. Claim rejections pose a major challenge among NHIF-accredited HFs that provide healthcare services to clients [1,4,7]. Claim rejection is a commonly occurring phenomenon not only in Tanzania but across the globe. It affects and disrupts service provision in different ways. Even with high experience in claims, rejections of claims occur with all providers [8,9]. Health insurance companies are known to reject around 14% of claims in healthcare, totaling up to USD 262 billion annually [8,10]. In addition, the denial of claims puts cost burdens, such as that of cost recovery (i.e., identifying reasons for the denial and errors, correcting the errors, and resubmitting them for reimbursement), on the healthcare providers [10].

Rejection rates from NHIF were reported to have increased significantly in the fiscal year 2021–2022, as opposed to the preceding year 2021–2020 [4]. This increase in rejection claims coincides with worries about a potential collapse of the NHIF. Furthermore, the NHIF has been developing stringent payment packages for multiple hospitals withdrawing from the scheme [11]. Hospitals in Mwanza, both public and private, are facing a major problem resulting from the high number of rejected claims, which is indicative of a complicated web of interrelated elements impeding the prompt payment of medical services [5]. Rejected claims disrupt the financial stability of HFs, delay patient care, and undermine the overall quality of healthcare services. This is because the rejected claims leave HFs with debts, problems in supporting their staff, and an inability to recoup funds needed for the continuum of care [12,13]. The issue of the rejection of claims can pose a significant risk, especially at a time when Tanzania is moving towards Universal Health Insurance.

While Tanzania is working on the development of universal health insurance coverage, the main provider of insurance, the NHIF, is struggling. There has also been a rise in claim rejections by the NHIF, and threats by health providers to boycott the NHIF [3,4,11,14]. Therefore, to optimize the NHIF’s operations and ensure the financial sustainability of healthcare institutions, there is a need to comprehend the patterns involved and investigate the hospitals’ coping mechanisms for claim rejection [5,8,15]. Studies suggest that understanding the patterns in the rejected claims will help facilities design appropriate investments to reduce rejected claims and allow for cost savings [2,4,16]. To this end, this study aimed to determine the patterns and mitigation strategies for claim rejection among health facilities providing services for the NHIF in Nyamagana district, Mwanza City.

In the next sections, we provide details of the methods and procedures used in the study; in addition, we present the findings, followed by a discussion and conclusion with a description of the limitations of the study.

## 2. Materials and Methods

This part outlines the research methodology used in this study to investigate the patterns of rejected claims and how hospitals cope with these rejections. It provides a detailed account of the research design, data collection methods, and data analysis techniques employed to address the research questions.

A mixed-methods approach combining quantitative surveys and qualitative interviews was chosen to gain a comprehensive understanding of the phenomenon. A cross-sectional study design was employed to capture quantitative data on the patterns of the rejections, and a qualitative section explored the coping strategies employed among healthcare facilities providing services for the NHIF in Mwanza. The use of these mixed methods allowed for triangulation of information [17,18].

This study was conducted in NHIF-accredited hospitals that provide healthcare services to NHIF clients in Mwanza City. Mwanza City is located in Northwestern Tanzania, along the shore of Lake Victoria, and consists of one district, namely, Nyamagana. Mwanza city has a population of around 426,154 people [19]. The city has 58 NHIF-accredited hospitals that provide healthcare services to NHIF beneficiaries, including zonal referral hospitals (2), regional referral hospitals (4), district hospitals (7), health centers (13), dispensaries (25), and polyclinics (7). The city of Mwanza is the second largest and second most-populated city in Tanzania, with both urban and rural settings. Its health system is representative of both urban and rural systems in the country.

Based on Yamane Taro’s formula, only 46 HFs were included in this study from a fixed number of 58 HFs in Mwanza [20,21]. HFs were categorized based on care level and ownership, and a category list was used as a sampling frame to ensure representation of each level. The first facility was selected by lottery and every other facility was selected after a Kth value determined by the number of facilities in the category. Data was collected using a checklist, entered into a computer using Epi-data version 3.1 (CDC, Atlanta, GA, USA), and analyzed using STATA version 15 (College Station, TX, USA). Data was summarized in the form of proportions and frequent tables for categorical variables. Various factors such as the independent variables (type of healthcare facilities, number of claims submitted, mode of claim submission (i.e., manually or electronically), reasons for claim rejection, coping), and dependent variables (rejected claims) in the categorical variables were explored. The chi-square test was used to test for the significance of the association between different patterns of rejections and different levels of health facilities. A *p*-value of <0.05 was considered to constitute a statistically significant difference.

Key informants included medical officers in charge/facility managers, quality assurance officers, and claim officers who were recruited for the study until saturation was reached [22,23,24]. Research assistants with a background in social sciences conducted interviews and recorded them using tape recorders. Interviews were guided by a prepared interview guide. All audio records were transcribed verbatim and then translated from Swahili, while expert translators ensured that the message was preserved. The following steps entailed reading the transcripts several times to gain a deeper knowledge of the material to identify patterns, topics, and areas of interest. A hybrid coding method (deductive and inductive coding) was used to achieve complete coding. This included segmenting data and ensuring it was categorized and labeled with applicable codes. Coding was carried out by two people for quality control purposes, using qualitative data analysis software (NVIVO 12). The creation of themes followed the coding. Commonalities and linkages between codes were recognized, and related codes were grouped to form larger themes. Two social scientists repeatedly analyzed the major themes’ categories to ensure accurate classification. Thematic coding helped the two social scientists to conduct a grounded analysis. This hybrid technique was employed to guide the advancement of both intentional and emergent themes. Data was analyzed using specific, methodical, and reproducible techniques, ensuring that the findings were validated and trustworthy and that the conclusions derived were evidence-based [25]. As a prerequisite for conclusions to be independently and objectively validated, we cited actual data in verbatim quotes from respondents. The age, gender, occupation, and settings of the study informants were listed at the end of each quotation.

## 3. Results

### 3.1. Facility and Key Informant Characteristics

This study included 46 HFs from the Nyamagana district of the city of Mwanza. Both public and private health facilities were included. Only 2.17% (n = 1) of hospitals submitted claims online, while 97.83% (n = 45) of HFs were paper-based. The enrolment included one zonal hospital, two regional referral hospitals, eleven health centers, four polyclinics, and twenty-five dispensaries. The key informant interviews included 10 participants with varied roles, including four hospital managers, two quality assurance officers, and four claimant officers, all of whom were responsible for claim submissions in their respective facilities. Most participants were aged 30–39, of male gender, with the majority holding a bachelor’s degree. Characteristics of informants are detailed in Table 1.

### 3.2. Patterns of Claim Rejection 

This study revealed a significant variation between facility types, as illustrated in Table 2 below. Zonal hospitals showed the highest average rejections per claim of 1.21, likely due to their handling of more complex cases and a higher volume of claims with more than one item, which increases the likelihood of one claim having multiple item rejections. In contrast, polyclinics, district hospitals, and health centers showed a moderate rejection average of items per claim of 0.39, 0.34, and 0.32, respectively, with regional referral hospitals having the lowest average (Table 2).

The findings revealed significant patterns of claim rejection across different HF levels, with distinct reasons emerging as dominant at various levels, as illustrated in Table 3 below. Non-adherence to standard treatment guidelines (STG) was most prevalent at polyclinics, accounting for 17.19% (133 items), followed by regional referral hospitals at 11.86% (666 items), while it was less common at zonal health facilities at 0.78% (1371 items). Overutilization of services was most significant at district and zonal facilities, at 31.79% (1615 items) and 26.04% (45,499 items), followed by health centers at 24.27% (431). Regional hospitals and polyclinics reported lower overutilization rates, at 13.98% (785 items) and 12.52% (268 items), respectively.

Attendance date anomalies occurred most frequently at dispensaries, at 14.8%, (201 items) and polyclinics at 14.11% (302), followed by health centers, which accounted for 11.49% (204); district facilities and regional centres reported 7.95% (404) and 7.87% (442), respectively, while zonal health facilities exhibited lower rates, at 0.12% (206). Improper dosage quantity was most frequently observed at the polyclinic and regional facilities, at 8.03% (172) and 7.85 (441), respectively, and least commonly at the zonal hospital, at 2.66% (4645).

Non-adherence to NHIF pricing was highly prevalent in dispensaries 12.74%, (173) and much lower in district facilities 2.24% (114 items). Duplicate/double claiming and medicines/consumables already included in the package were observed significantly more often at the zonal facility, at 10.63% (18,570 items) and 8.41% (14,695), followed by regional facilities at 10.08% (566 items) and 2.24% (126 items), respectively. Non-indicated diagnoses were encountered more frequently at the dispensary level, at 7.36% (100 items), followed by regional facilities accounting for 6.73% (378 items), with the lowest at the zonal level of 0.89% (18,570 items).

Improper coding was observed more often at polyclinics, at 5.28% (113), and the least frequently at the zonal health facility, which accounted for 0.00% (2 items). Among facilities using a paper-based system, items not claimed in physical form were not seen at the zonal facility, but were significantly reported at dispensaries, at 4.12% (56), with the lowest frequency being at polyclinics, at 0.79% (17).

Other reasons for rejection, including missing patient information/signature, observed more often at health centers (8.39% (149)), and wrong entry of items at polyclinics (9.29% (199)), followed a similar pattern of being more frequent at lower facilities compared to higher-level facilities like zonal facilities. Consultation charges not applicable to visit type were only seen at the zonal facilities, at 42.66% (74,499 items). The category “Other” accounted for a substantial portion of rejections at regional facilities, representing 22.68% (1274), and at zonal facilities accounting for the lowest portion of 4.75% (8299 items).

In the Pearson Chi-square test, the results indicated that there is a significant association between the level of the healthcare facility and the number of rejected claim items (*p* < 0.001). This suggests that the variation in the number of rejected items is not random, but is significantly related to the type or level of the facility.

Certain rejection patterns, such as overutilization and non-adherence to Standard Treatment Guidelines (STG), encompass multiple components. These include issues related to investigations, medications, and other services. Table 4 and Table 5 below illustrate these various components. At dispensaries, health centers, polyclinics, and regional and zonal centres, overutilization rejection due to drugs was high compared to overutilization due to investigational tests: 58.80% (137), 56.15% (242), 57.46% (154), 59.49% (467), and 71.31% (32,445), respectively. In contrast, at the district level, overutilization due to investigational tests was more significant, at 62.54% (1010), as compared to overutilization due to drugs, which was 35.85% (579). Polypharmacy was also seen at all levels of facilities except at the regional level, and the highest was seen at health centers, at 4.41% (19).

Nonadherence to STG with drug components was more frequently observed at the dispensary (80.91% (89)), health centers (69.17% (92)), polyclinics (62.23% (229)), district (59.09% (221)) and zonal levels (88.99% (1220)), with an exception for the regional level, which had more rejections of non-adherence to STG due to investigational tests, at 50.30% (335), compared to non-adherence to STG due to drug items, at 49.70% (331). One health facility, a Zonal Referral Hospital (2.17%), submitted claims online, while 97.83% (n = 45) of health facilities were paper-based.

Regarding the qualitative part of the analysis, the key informant interviews included 10 participants with varied roles, including four hospital managers, two quality assurance officers, and four claimant officers, all of whom were responsible for claim submissions in their respective facilities. Most participants were aged 30–39, of male gender, with the majority holding a bachelor’s degree. Characteristics of informants are detailed in Table 1.

### 3.3. Rejected Claims and Mitigation Strategies

The qualitative findings revealed that claim rejections can be broadly categorized based on whether they originate from NHIF policies or practices within the health facilities, or not. This was based on information gained from key informant interviews with claimant officers/hospital managers.

#### 3.3.1. NHIF Policies

NHIF policies and their changes constantly lead to rejected claims. These policies are usually meant for cost-cutting and sometimes for the strict control of the use of services. However, because the policies are inflexible and cannot deal with case-by-case situations, they cause a lot of rejected claims. NHIF policies tend to be strict and do not allow real-time flexibility, as multiple factors can cause changes to claims. For example, one participant acknowledged:

“*Sometimes, delays in receiving updated price information cause us to create bills with old prices which later on lead to rejection.*”KII with a 34-year-old female health facility manager.

Another participant was even more straightforward in stating the impacts of price changes in her facility:

“*The price may change between months and without regard for any claims that were not sent before; for example, in-patient claims, due to online protocol, we will be compelled to alter the item pricing, and those sent with the previous price will be rejected automatically once the claims are submitted in their system for reimbursement.*”KII with a 42-year-old male quality assurance officer.

The policy does not only affect claims due to pricing. The policy also affects the physicians’ decision on having to choose a higher-priced drug for clinical reasons. Participants revealed that some rejected claims originate from policies that have been established by the NHIF. Hospitals are required to follow provided STGs for managing patients. However, the NHIF rejects the prescription of high-cost drugs in comparison to those with a low price to the clients, despite being in Tanzania’s standard treatment guidelines/NEMLIT and their contracts. This can happen with drugs such as ferrous folic acid capsules versus ferrous folic tablets. This can be explained by the fact that NHIF policies mostly aim at cost-effectiveness without considering the logic behind the clinician’s choice of one drug over another. A respondent revealed:

“*NHIF will force you to choose the least effective drug driven by their cost-cutting policies which do not always account for potential side effects or the clinician’s rationale for choosing one medication over another. And such drugs will be rejected.*”KII with a 49-year-old male Facility Manager.

Other claim rejections can occur due to policy changes and lead to the rejection of all previous claims that were not considered under such a change. A 31-year-old female claimant described such an atypical situation as follows:

“*Physiotherapy services were initially exempt from requiring prior approval. However, at a certain point, the NHIF introduced a new control measure, mandating prior approval for these services. This change led to the rejection of all physiotherapy claims submitted by our health facility with a reason of no invalid approval letter.*”

#### 3.3.2. Health Facility Practices

Rejections due to health facility practices are primarily internal and stem from operational flaws, insufficient training, and communication problems, such as missing details of services/procedures carried out when submitting the forms, no/invalid authorization number, missing details of services claimed to be provided after verification of claims, missing physical forms, item not claimed in the physical form, missing clinician/patient’s signature, claiming unissued services, attendance date anomalies, alteration of admission dates, absent or improper dosage and quantities, investigations/medicines/medical consumables already included in the package, long hospital admission without notification, non-indicated diagnosis, and wrong entry of the item. One informant noted:

“*Doctors don’t write enough notes to justify the treatment they gave to patients, so it leads to deductions due to the lack of detailed history of the patient and bad writing of insufficient codes.*”KII with a 40-year-old male Health Facility Manager.

Furthermore, due to the reliance on manual processes, a significant portion of claim rejections can be attributed to the practices within HFs. Common issues include errors in documentation, loss of essential paperwork, and incomplete or inaccurate claim submissions. A participant reported:

“*We frequently experience issues such as the loss of crucial documents, including NHIF approval letters and claim forms. Additionally, forms for long hospital stays are often misplaced, and documentation errors are more common such as invalid authorization numbers.*”KII with a 34-year-old female Health Facility Manager.

Other facility practices include the late submission of claims, which may result from inefficiencies in internal processes, delays in documentation, and poor coordination. As a result, claims are often not submitted due to past deadlines. For example, a 49-year-old male facility in charge explained:

“*Certain forms are sometimes discovered weeks after claims have been submitted, resulting in automatic rejection by the NHIF office, as these forms are no longer accepted.*”

### 3.4. Mitigation Strategies Utilized to Avoid Claim Rejections

The key informant interviews revealed a variety of measures employed by HFs when confronted with claim rejections from the NHIF. Many participants reported an immediate response in prioritizing rejected claims by first verifying the errors, correcting them, and ensuring they do not recur in other claim submissions. This was the dominant approach reported by many participants, which involved conducting thorough internal audits and reviews of rejected claims. Several key informants described this process. For example, a respondent reported:

“*Upon receiving a rejection, the facility’s initial step is to prioritize the rejected claims, and rejections are immediately flagged and evaluated to determine the reason for the rejections. This involves verifying the claim details against the specific rejection criteria provided by NHIF, such as non-adherence to Standard Treatment Guidelines (STG) or errors in documentation.*”KII with a 33-year-old male Insurance Claimant.

This was also reported by another participant, who stated:

“*After receiving the payment summary, our unit convenes to review all rejections. We go through each case, carefully examining the reasons for rejection.*”KII with a 39-year-old male Quality Assurance Officer.

Furthermore, all participants agreed that they communicated regularly with the NHIF to get an understanding of the rejected claims to obtain clarity and improve future practices. A participant reports:

“*Whenever a new pattern of rejection occurs, our first step is to contact the NHIF office to understand the specific reason behind the rejection. This helps us to avoid similar issues in future submissions.*”KII with a 33-year-old male Claimant Officer.

Participants also highlighted the importance of establishing clear, proactive communication channels with the NHIF. Regular meetings and direct consultations with NHIF representatives were cited as effective ways to gain insights into common rejection reasons and how to avoid them. As one informant explained:

“*We have regular meetings with NHIF to discuss rejection trends. This proactive engagement helps us stay ahead and reduce the chances of future rejections.*”KII with a 39-year-old male Quality Assurance Officer.

Also, communication with the NHIF regarding difficulties in sending claims due to network/internet breakdowns to avoid rejections due to time delay was reported to smoothen the process of eliminating late submission rejections.

Training of staff involved in the claim process was implemented by all facilities. This is particularly crucial in cases where rejections are due to common errors such as incorrect coding or incomplete documentation. One participant stated:

“*After we noticed a pattern in rejections due to coding errors, we organized refresher training sessions for our billing staff and medical personnel. This has significantly reduced the number of rejections.*”KII with a 37-year-old male Facility Manager.

All participants emphasized the importance of regular training sessions for staff involved in the claim submission process and other staff. Continuous professional development education regarding adherence to standard treatment guidelines, proper code filling (provisional diagnosis will be based on the investigations ordered and the final diagnosis will be for prescribed medications), and adhering to NHIF guidelines and policies was considered very crucial in making sure they are abided and followed thoroughly. According to one participant:

“*Regular training is essential because healthcare policies and NHIF guidelines frequently change. We ensure our team is up to date to minimize errors that lead to rejections.*”KII with a 30-year-old male Claimant Officer.

In addition, other participants reported:

“*Training on adherence to NHIF standards and following standard treatment guidelines is an ongoing process at my hospital since we acquire new personnel here and there.*”KII with a 42-year-old Quality Assurance Officer.

The NHIF provides such training to health facilities; this was acknowledged by one participant who said:

“*We have an NHIF coordinator who supports us by explaining and training on the reasons for claim rejections, as well as guiding us on the appropriate steps to effectively address these issues.*”KII with a 34-year-old female Facility Manager.

Reviewing errors in the claims before submission is also carried out in health facilities. Before submitting the claims for reimbursement, the forms are thoroughly checked and evaluated for missing details that might facilitate rejections and for duplicate items. One participant shared:

“*Sometimes, the rejection is due to missing documentation which might be due to incomplete codes or doctors’ signature in which the forms are taken back to the person responsible and make sure he/she fills appropriately.*”KII with a 33-year-old male Claimant Officer.

Efforts to combat fraud are put in place to ensure all people adhere to protocols. Sometimes clinicians tend to sympathize with the client when they find out the NHIF package does not cover the patient’s procedure/surgery. Moreover, some may forge and bill the procedure with higher costs than the actual procedure carried out or claim some surgeries not undertaken in reality. Fraud is found after intensive verification by NHIF officers, known as the anti-fraud team, with the involvement of patients without the presence of a health facility representative. On the mitigation strategy against fraudulent activities, one respondent reported:

“*We have implemented a stronger penalty for people discovered to engage in fraudulent activity, including fines.*”KII with a 39-year-old male Quality Assurance Officer.

Another respondent added:

“*We have established a Zero Tolerance Policy: whereby we implemented and executed a zero-tolerance policy against fraud, emphasizing that any found fraud would result in severe repercussions. In addition, there is continuous professional development training among staff and educating patients about fraud and encourage them to report suspicious practices such as being billed for services they didn’t receive.*”KII with a 40-year-old male Facility Manager.

Lastly, participants reported the use of technological solutions as a mitigation strategy. An online claim submission system is part of the response strategy against rejected claims to minimize human errors and streamline the submission process. They reported that the online system allowed for a quicker rectification of errors and more efficient submissions. One informant mentioned:

“*The online system is a game-changer for us. It has helped in eliminating rejections such as invalid authorization numbers, missing health facility seals, missing doctor’s signatures, and patient signatures. Before the submission the system will force such things to be in the correct order, minimizing the impact of the rejections.*”KII with 37-year-old Insurance Claimant.

It was, however, reported that slow or unreliable systems, frequent downtimes, and a lack of technical help from health insurance providers all impeded the online submission process.

## 4. Discussion

This study aimed to investigate the patterns, magnitude, and mitigation strategies of rejected NHIF claims across different healthcare facilities in Mwanza City, Tanzania. It provided a comprehensive analysis, offering a deeper understanding of the underlying causes of claim rejections and the strategies used to address them. A key novelty of the study was its focus on the relative potential influence of facility levels, submission methods, and NHIF policies, revealing distinct rejection patterns that vary across institutions.

The results of this study demonstrated a significant variability in claim rejection rates across different levels of healthcare facilities in Mwanza City. A zonal hospital displayed the highest average rejection of items per claim of 1.21, while others had significantly lower rates of items rejected by claims, with the regional facilities having the lowest average of 0.21. These results highlight that a higher-level facility, such as a zonal referral facility, tends to face greater challenges in rejections and managing claims, with higher rejection rates per item—likely owing to their handling of more complex cases and a higher volume of claims with more than one item, which raises the risk of one claim having multiple items rejected. The patterns of rejection are different across facilities because of service levels and types of services. Similar findings of a United States of America (USA) study showed that some services are more commonly rejected than others, leading to revenue loss in different institutions [26].

The study revealed important and sometimes distinct reasons for rejected claims among the different levels of HFs, but most appeared to stem from genuine errors rather than intentional attempts to cheat the system. These errors were often unintentional and resulted from a combination of factors, including clerical mistakes, misinterpretation of NHIF guidelines, and inadequate staff training. Clerical errors, such as incorrect patient information or service dates, were common, as were mistakes related to coding and documentation. The most common rejection patterns included overutilization of medications, improper dosage quantities, non-adherence to standard treatment guidelines (STGs), and NHIF pricing, not indicated by diagnosis, improper coding, double claiming, and attendance data anomalies. These rejections occurred regularly at all levels of healthcare facilities, with minor fluctuations in frequency depending on the kind of facility. These findings are consistent with previous research, which showed that claim rejections are common in health insurance systems due to a variety of variables such as operational inefficiencies within health institutions [27,28]. One study reported that errors in claims were the dominant factor for rejection, including coding errors (treatment–diagnosis mismatch), over-prescription/inappropriate prescription, overutilization of medicines, and treatment or prescription beyond provider accreditation [29]. Our findings corroborate the 2020/21 CAG report, which highlighted that the reasons for rejections for Muhimbili National Hospital and Jakaya Kikwete Cardiac Institute included calculation errors, missing details of services claimed to be provided, improper coding, double claiming, proven cases of fraud, no/invalid NHIF approval letter, invalid/authorization number, no/invalid clinician or patient signature, non-adherence to STG, not in NHIF benefit package, overutilization of prescribed item, overprescribing, invalid/no seal of health facility on form 2C/2E, and long hospital admission without notification [30]. The study found that rejections due to non-adherence to Standard Treatment Guidelines (STG) and overutilization were primarily related to drugs across all health facilities. However, at the district level, rejections due to overutilization were also frequently attributed to investigations, and at the regional level, non-adherence investigation-related rejections remained predominant. Overutilization of medication in zonal referral hospitals could be explained by specialists considering that patients must have received partial treatments and medication at lower levels, hence the need for appropriate, focused patient management at the zonal hospital that requires more investigations and medications. Furthermore, specialists at higher levels feel the urge to demonstrate the power of choice rather than using the restricted STGs. On the other hand, overutilization of diagnosis was found to be common in district-level private hospitals, pointing to the possibility of an increase in the income of the hospitals [31]. Rejected claims reduce the expected revenue of healthcare providers, leading to cash flow problems. Facilities depend on reimbursements from the NHIF to cover the costs of services rendered. When claims are rejected, these facilities may face budget constraints, limiting their ability to provide timely services, procure medications, or maintain staff salaries. These findings align with the study carried out in Ghana, which reported that delays in claim reimbursement by the National Health Insurance Authority hurt the financial management of healthcare facilities [32].

However, some of the rejections were also attributable to the strict documentation and compliance requirements mandated by the NHIF. The complexity of NHIF policy guidelines, as well as frequent policy revisions, were highlighted as important factors in claim rejections due to non-adherence to NHIF pricing, which frequently contributes to confusion and errors in claim filing. Insurance regulations and legislation change over time and affect the claims procedure. Health facilities must adjust to these changes, which can be complicated and time-consuming.

The data also found that the majority of rejections were the results of the internal processes inside health facilities. Operational inefficiencies and poor documentation standards were major contributors to claim denials. These findings are congruent with several studies that have found several issues in health insurance claim management across different countries. In Ghana, administrative capacity concerns, technical challenges, and human resource limits all contributed to delays in claims submission and processing under the National Health Insurance Scheme [27]. In Indonesia, Tanjung Pura Hospital discovered that 2.9% of claims were rejected due to administrative verification problems, such as missing signatures and inadequate documents [33]. Another study at the same hospital supported these findings, citing a lack of thorough evaluation and the absence of standard operating procedures as contributory causes. In Pontianak, Indonesia, claim rejections were linked to administrative errors, coding conflicts, and perceptual disparities between hospital coders and insurance verifiers [34]. To reduce claim rejections and delays, these studies emphasize the importance of increased administrative capacity, standardized procedures, and enhanced communication between healthcare providers and insurers.

The analysis also indicates that as more health facilities submit claims for reimbursement, the greater the likelihood of encountering rejections. As more health facilities submit claims, the complexity and scrutiny of those claims also increase, leading to several rejection grounds. This trend is concerning, as it indicates that, despite the increasing adoption of health insurance, many facilities struggle to meet the submission standards required by NHIF resulting in financial strain. This increase could be attributed to several factors including insufficient staff training. Additionally, the increase in claims might overwhelm the existing administrative capacity of both the healthcare facilities and NHIF, leading to more frequent errors and oversights that trigger rejections. A similar observation was made by a study conducted in Ghana, which showed an increase in claims for reimbursement over the period 2011–2013 consistent with the proportion of rejected claims, which also increased from 0.9 to 3.6 over the same period [35]. In addition, Tanzania’s NHIF Annual Performance Report and Audited Financial Statements from 2015/16 to 2017/18 revealed a considerable increase in rejections of claims, as there was an increase in the amount claimed from accredited healthcare facilities across the country [30].

A particularly promising finding from this study is the role of technology; in particular, online claim submission systems can play a major role in reducing the rate of rejections, especially for rejections due to human errors such as invalid/no authorization number, wrong entry of item, invalid/no seal, missing/no clinician signature, missing patient’s information, missing physical claim form, and an item not included in the physical form. While the data showed that only 2% of health facilities in the study use electronic submission, it is important to note that this small proportion does not diminish the impact of the electronic system on the overall claim submission performance. The one facility utilizing electronic submission was a zonal hospital, which, by design, processes a significantly higher volume of claims compared to lower-level facilities such as dispensaries, health centers, and regional or district hospitals. Despite being only one among many facilities, its contribution to the total claim volume was disproportionately large, accounting for a major portion of the claims processed within the entire system. The efficiency gains from using an electronic system are particularly noteworthy, reducing clerical errors, such as missing signatures, incorrect card numbers, missing physical forms, missing items in physical forms, and authorization issues, which are more prevalent in paper-based systems. This efficiency is crucial in a facility managing such a high volume of claims. Hence, this provides a clear demonstration of how technology can streamline the claims process, especially in high-volume, high-level healthcare settings. This finding was supported by a study that compared paper-based claims to digital health technology and concluded that digital technology minimizes health insurance claim rejections [16]. Facilities that have adopted online submission systems are acknowledged to have successfully eliminated such rejections, while other health facilities relying on paper are still struggling with such types of rejections. This reduction in rejections is likely due to the automation and standardization that online systems allow by minimizing human errors. Online systems often include built-in checks that alert submitters to common errors before the claim is finalized. This proactive error-checking mechanism can prevent many of the issues that lead to rejections, making the claims process more efficient and reducing the administrative burden of health care providers. A study carried out in Ghana concluded that the highest rejection rates were seen among paper-processing claims, as compared to online claims, which also improve the efficiency of the National Health Insurance Scheme [36]. The use of electronic claims aimed to reduce human intervention and errors in healthcare provider claims processing.

In this study, the majority of key informants were unsure about the specific details of their claim’s rejections—some claim rejections had different meanings as compared to the reason assigned. These findings align with a study that reported serious issues including high rates of non-reimbursement of claims due to a lack of feedback about errors, and a lack of clarity on claims reporting procedures [37]. This observation highlights a distinctive aspect of NHIF-rejected claims that implies that communication breakdowns and insufficient clarity in the rejection process may cause misunderstanding among healthcare practitioners, potentially contributing to greater rates of unsettled claims and financial hardship on health facilities. This lack of feedback and clear rules impedes providers’ capacity to handle claim difficulties, while also undermining trust in the NHIF system as a dependable source of reimbursement. Generally, these findings highlight the significant implications of both policy and practice; there is a clear need for the NHIF to revise and enhance its guidelines and training programs to address the new patterns of rejections that have emerged and strengthen communication with healthcare providers once they have made changes regarding their services provision to reduce the burden of rejected claims among health care facilities. For healthcare facilities, the findings underscore the importance of investing in staff training, especially in areas of claim submission and compliance with NHIF guidelines. Facilities that have not yet adopted an online submission system should consider doing so as a means of reducing rejection rates and improving their financial sustainability.

The strength of this study is the use of a mixed-method design, which provided a comprehensive analysis of the problem. The quantitative aspect offered a statistical insight into the patterns of rejected claims, while the qualitative component delved deeper into the reasons behind these patterns through interviews with key stakeholders. This dual approach ensured a more complete and holistic understanding of the issue, balancing numerical data with personal and institutional perspectives. However, the study had some limitations, such as the fact that it did not utilize econometric operations to address various issues such as the monotonicity of presentations, and only relied on the trustworthiness of the data findings to derive the observation. Furthermore, the study focused primarily on the perspective of claimant officers from health facilities, without considering the insights of NHIF administrators or policymakers, a potential source for information bias. This narrow focus might have overlooked broader systemic issues or alternative viewpoints that could have provided a more comprehensive understanding of potential challenges. However, the tools were designed to be objective and focus on eliciting information regarding the patterns and coping strategies of health facilities

## 5. Conclusions

Insurance claims rejections present a major challenge to both private and public healthcare facilities due to the financial strain encountered, hence limiting the facilities’ capacity to sustain overall operations. The findings from this study revealed that the most frequently occurring rejection patterns include overutilization, non-adherence to STG/NHIF pricing, improper dosage quantity, double claiming, improper coding of diseases, not indicated by diagnosis, wrong entry of the item, and attendance data anomalies, which were observed across all HFs. The use of an electronic claim submission system was effective in reducing the rejection of claims and should be adopted in all HFs. In addition, hospitals should consider the use of efficient electronic systems that will detect errors in a timely manner during patient management to ensure timely corrective measures. However, rejection rates lacked precision, as they were recorded by items rather than the number of claim folios submitted, making it difficult to fully assess the impact on health facilities. A detailed payment summary, indicating the items involved and the reasons for rejection, would provide greater transparency and clarity. We recommend studies involving NHIF policymakers and a longitudinal study to track the impact of implemented mitigation strategies over time.

## Figures and Tables

**Table 1 healthcare-13-00320-t001:** Facility and Key Informant Characteristics.

Characteristic	Frequency	Percentage
Facility Level		
Zonal	1	2
Regional	2	4
District	3	7
Health Centre	11	24
Dispensary	25	54
Polyclinic	4	9
Facility Ownership		
Public	27	59
Private	19	41
Mode of Insurance Claims Submission		
Online	1	2
Paper Based	45	98
Position		
Medical officer in charge/Manager	4	40
Quality assurance officer	2	20
Claim officer	4	40
Age		
30–39	8	80
40–49	2	20
Gender		
Male	7	70
Female	3	30
Education background		
Diploma	3	30
Bachelor	7	70

**Table 2 healthcare-13-00320-t002:** Average of Rejected Items per Claims for Six Months in Nyamagana District Health Facilities.

Level of Facility	Total Claims Past 6 Months	Rejected Items	Average Rejected Item per Claim
Dispensary	5039	1358	0.27
Health center	5469	1776	0.32
Polyclinic	5553	2141	0.39
District	14,787	5081	0.34
Regional referral	27,065	5617	0.21
Zonal	144,860	174,696	1.21

**Table 3 healthcare-13-00320-t003:** Patterns and Magnitude of Rejected Claims.

Variable	Dispensaries (n = 1358)	Health Centre (n = 1776)	Polyclinic (n = 2141)	District (n = 5081)	Regional (n = 5617)	Zonal (n = 174,696)	*p*-Value
Non-adherence to STG	110 (8.1%)	133 (7.49%)	368 (17.19%)	374 (7.36%)	666 (11.86%)	1371 (0.78%)	<0.001
Overutilization	223 (16.42%)	431 (24.27%)	268 (12.52%)	1615 (31.79%)	785 (13.98%)	45,499 (26.04%)	<0.001
Attendance data anomalies	201 (14.8%)	204 (11.49%)	302 (14.11%)	404 (7.95%)	442 (7.87%)	206 (0.12%)	<0.001
Improper dosage quantity	98 (7.22%)	119 (6.70%)	172 (8.03%)	380 (7.48%)	441 (7.85%)	4645 (2.66%)	<0.001
Non-Adherence to NHIF pricing	173 (12.74%)	70 (3.94%)	74 (3.46%)	114 (2.24%)	223 (3.97%)	4938 (2.83%)	<0.001
Duplicate/Double-claiming	57 (4.20%)	57 (3.21%)	62 (2.90%)	483 (9.51%)	566 (10.08%)	18,570 (10.63%)	<0.001
Not indicated in the diagnosis	100 (7.36%)	118 (6.64%)	120 (5.60%)	146 (2.87%)	378 (6.73%)	1563 (0.89%)	<0.001
Wrong entry of Item	45 (3.31%)	81 (4.56%)	199 (9.29%)	176 (3.46%)	163 (2.90%)	409 (0.23%)	<0.001
Missing Patient information	64 (4.71%)	149 (8.39%)	90 (4.20%)	109 (2.15%)	190 (3.38%)	0 (0.00%)	<0.001
Consultation Charges Not applicable	0 (0.00%)	0 (0.00%)	0 (0.00%)	0 (0.00%)	0 (0.00%)	74,499 (42.64%)	<0.001
Already in package	5 (0.37%)	5 (0.28%)	34 (1.59%)	93 (1.83%)	126 (2.24%)	14,695 (8.41%)	<0.001
Improper coding	53 (3.90%)	61 (3.43%)	113 (5.28%)	217 (4.27%)	236 (4.20%)	2 (0.00%)	<0.001
Items not claimed in physical form	56 (4.12%)	30 (1.69%)	17 (0.79%)	150 (2.95%)	127 (2.26%)	0 (0.00%)	<0.001
Other *	173 (12.74%)	318 (17.91%)	322 (15.04%)	820 (16.14%)	1274 (22.68%)	8299 (4.75%)	<0.001

* Other includes invalid authorization number, no NHIF approval letter, long hospital admission without notification, missing physical form, verification of claims, alteration of admission dates, invalid seal, invalid clinician signature, proven cases of fraud, the substitution of services, wrong clinician consultations, not in NHIF package, missing details of surgery/procedure/service, calculation error, claiming unissued services, improper admission charges, and others. Color key: Green (0–4.99%) Yellow (5–9.99%) Orange (10–19.99%), Red (≥20%).

**Table 4 healthcare-13-00320-t004:** Description of overutilization Pattern.

Category	Dispensary (N = 233)	Health Center (N = 431)	Polyclinic (N = 268)	District (N = 1615)	Regional (N = 785)	Zonal (N = 45,499)
Test overutilization	79 (33.91%)	170 (39.44%)	109 (40.67%)	1010 (62.54%)	318 (40.51%)	12,821 (28.18%)
Item/drug overutilization	137 (58.80%)	242 (56.15%)	154 (57.46%)	579 (35.85%)	467 (59.49%)	32,445 (71.31%)
polypharmacy	7 (3.00%)	19 (4.41%)	5 (1.87%)	26 (1.61%)	0 (0.00%)	170 (0.37%)
Other *	0 (0.00%)	0 (0.00%)	0 (0.00%)	0 (0.00%)	0 (0.00%)	63 (0.14%)

* Other—overutilization due to dosage/quantities, overutilization of inclusive visits, overutilization of other services.

**Table 5 healthcare-13-00320-t005:** Description of Nonadherence to STG.

Category	Dispensary (N = 110)	Health Center (N = 133)	Polyclinic (N = 368)	District (N = 374)	Regional (N = 666)	Zonal (N = 1371)
Non-adherence investigation	21 (19.09%)	41 (30.83%)	139 (37.77%)	151 (40.37%)	335 (50.30%)	150 (10.94%)
Non-adherence item/drug	89 (80.91%)	92 (69.17%)	229 (62.23%)	221 (59.09%)	331 (49.70%)	1220 (88.99%)
Non-adherence consultation	0 (0.00%)	0 (0.00%)	0 (0.00%)	0 (0.00%)	0 (0.00%)	1 (0.07%)
Non-adherence procedure	0 (0.00%)	0 (0.00%)	0 (0.00%)	2 (0.53%)	0 (0.00%)	0 (0.00%)

## Data Availability

All data provided in this manuscript informs of tables, numbers, and quotations.

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
