# Peer review of "Patterns and Mitigation Strategies for Rejected Claims Among Health Facilities Providing Services for the National Health Insurance Fund in Mwanza, Tanzania"

_healthcare, 2025, doi:10.3390/healthcare13030320_

Round 1

Reviewer 1 Report

Comments and Suggestions for Authors

Failure to claim the health insurance facilities is a common problem in any economies. It depends on how the information plays a big role behind all sorts of transactions. The authors have tried to find the causes behind rejections of insurance claims in Mwanza of Tanzania. The study has used statistical tools for reaching the results and their analyses. The results are expected. It has some degrees of policy implications. However, the study has some lacunae in some of the areas as mentioned below and the rectifications/revisions of which may be leading to a good research outcome.

1.      The Abstract looks good but it should have been in a single paragraph with no sub-headings like Background, etc. so far as the template of the journal is concerned.

2.      The study has a brief introduction to the subject area. It mainly deals with the issues related to insurance claims in Tanzania in particular. Before vending towards insurance related issues the authors could have included two related topic into discussion. One, the causes like urbanization, pollution upon health cost, two, the global or other countries’ scenario of insurance claim rejections, etc.

3.      It is not clear about the motivations behind the study? All have no clear explanation; it is suggested to incorporate the same. Further, the study should categorically mention the Research Gaps to justify its contributions to the scientific literature as it used the data and results of so many other authors. Why Mwanza is selected? Is it that it consists of more than 75 % of total health insurance facilities of all the cities in the country, or whether it has a significant population etc.

4.      In the health insurance market in particular, there arises the moral hazard problems with asymmetric information. The author should have gone with examining whether any such behaviour was there behind the rejections of the claims.

5.      The authors are suggested to go for impact assessment of different factors for claim rejections through econometric techniques, like individual or pooled regressions.

6.      The study did not present any theoretical basics of the interrelationships between the two key indicators without having any model. It is suggested to form a basic model with at least linking the two variables in equational structure to give a break to the monotonicity in the presentations.

Major revision recommended

Author Response

Dear Reviewer,

We thank you for the time you took to review our work. We have addressed your concerns and how work looks better. We are open to working on more comments should you have any. Please find attached, point-by-point revisions.

Reviewer 2 Report

Comments and Suggestions for Authors

Review

Introduction:

    • Strengthen the contextual background by elaborating on why Mwanza was selected as a case study compared to other regions in Tanzania.
    • Briefly discuss the importance of NHIF in reducing financial barriers to healthcare to emphasize the significance of addressing claim rejections.

Methods:

    • Provide more detail about the sampling method used for selecting the 46 healthcare facilities. For example, was it stratified random sampling, or were facilities purposively selected?

Results:

    • Table 2 includes data on rejected claims but could be enhanced by highlighting trends over time, if available, to indicate whether claim rejection patterns are improving or worsening.

Discussion:

    • The discussion section links the findings to previous studies effectively but could include more reflection on why overutilization is more prevalent at certain facility levels.

Limitations:

    • Discuss the potential for bias in qualitative responses, particularly given the reliance on self-reported practices by facility managers and claim officers.

Conclusion:

    • Emphasize the potential of electronic claim submission systems in reducing rejection rates and recommend steps for broader implementation in Tanzania.
    • Suggest further research to include NHIF policymakers and a longitudinal study to track the impact of implemented mitigation strategies over time.

Recommendation: Accept with minor revision.

Author Response

Dear Reviewer,

Thank you for your comments. We have worked on them to improve the manuscript. We are open to working on more suggestions should you have any. Please find point by point review of the manuscript. 

Reviewer 3 Report

Comments and Suggestions for Authors

see attached.

Author Response

Dear Reviewer,

Thank you for the time to review the manuscript. We have addressed the comments and improved our work. We are open to working on more comments should you have any.

Reviewer 4 Report

Comments and Suggestions for Authors

This study examines the patterns, magnitude, and mitigation strategies of NHIF rejected claims in various healthcare facilities in Mwanza City, Tanzania. The study, conducted between July and August 2024, used a mixed approach, combining quantitative data (from August 2023 to January 2024) from 46 healthcare facilities and qualitative interviews with key informants.

Among the strengths: 

·      Mixed approach: The use of both quantitative and qualitative research methods allows for a more comprehensive understanding of the problem.

·      Focus on different levels of facilities: The study includes a variety of healthcare facilities, including zonal hospitals, regional referral hospitals, district hospitals, health centers, dispensaries, and polyclinics. This enables a comparative analysis of rejection patterns across different levels of the healthcare system.

·      Identification of specific rejection patterns: The study highlights the most common reasons for rejection, such as excessive use of drugs, incorrect dosages, non-compliance with standard therapeutic guidelines (STG) and NHIF pricing, unindicated diagnoses, coding errors, duplicate claims, and anomalies in attendance dates.

·      Original contribution: The analysis shows how different levels of healthcare facilities influence rejection rates, providing useful insights for tailoring intervention strategies.

However, the structure of the study requires significant additions and modifications to be considered for publication:

1.     Introduction: At the end, include an explanation of the structure of the subsequent sections of the study.

2.     Background: Include a section appropriately explaining the state of the art, supported by relevant literature.

3.     Insert a subsection “Related Studies” that explains existing studies in the literature and any improvements in results/methodology achieved compared to these.

4.     Methodology: Insert an initial paragraph explaining the methodology used, potentially including a graphical workflow to provide an immediate and intuitive explanation (aligned with the modifications to be made).

5.     The study highlights correlations between the level of the healthcare facility and the number of items in rejected claims. Pearson’s chi-square test demonstrated a significant association between these variables (chi-square (45) = 790, p < 0.001). This indicates that the variability in the number of rejected items is not random but is significantly correlated with the type or level of facility. The study is limited to highlighting correlations between variables. For instance, machine learning techniques could be used to deepen data analysis and identify more complex patterns. Classification algorithms could be used to predict the likelihood of claim rejection based on the characteristics of the healthcare facility and the details of the claim itself. Alternatively, clustering algorithms could be used to identify groups of healthcare facilities with similar rejection patterns.

6.     Conclusions: These should obviously be appropriately aligned with all modifications made, including relevant limitations and implications.

Author Response

Dear Reviewer,

We thank you for the time to look at our work. We have addressed your comments and improved our work. We are open to working on more comments should you have any.

Round 2

Reviewer 1 Report

Comments and Suggestions for Authors

Good revision

Reviewer 4 Report

Comments and Suggestions for Authors

Dear Authors,

We appreciate the effort you put into incorporating our suggestions. The revisions have improved the manuscript, and we believe the article provides a valuable contribution.

We wish you the best of luck with the editorial process.